# Ceruloplasmin Interferes with the Assessment of Blood Lipid Hydroperoxide Content in Small Ruminants

**DOI:** 10.3390/antiox12030701

**Published:** 2023-03-12

**Authors:** Stefano Cecchini Gualandi, Raffaele Boni

**Affiliations:** Department of Sciences, University of Basilicata, Via dell’Ateneo Lucano 10, 85100 Potenza, Italy

**Keywords:** ceruloplasmin oxidase activity, lipid hydroperoxide (LOOH) levels, reactive oxygen metabolites (ROMs), sodium azide, total oxidant status (TOS)

## Abstract

Simple and inexpensive analytical methods for assessing redox balance in biological matrixes are widely used in animal and human diagnostics. Two of them, reactive oxygen metabolites (ROMs) and total oxidant status (TOS), evaluate the lipid hydroperoxide (LOOH) content of the sample and are based on iron-mediated mechanisms. However, these tests provide uncorrelated results. In this study, we compared these two tests in the blood serum of goat kids and lambs, together with an evaluation of ceruloplasmin (CP) oxidase activity. No significant correlation was found between ROMs and TOS, or between TOS and CP oxidase activity, in either species. Conversely, ROMs and CP oxidase activity were highly correlated in both kid and lamb samples (*p* < 0.001). A significant progressive reduction in the analytical signal in the ROMs assay was observed when sodium azide, an effective CP inhibitor, was added to the samples before the assay (*p* < 0.001). This decrease was related to sodium azide concentration (*p* < 0.01) and was not found when sodium azide was added at the same concentrations in the TOS assay. These findings suggest that ROMs, unlike TOS, may be affected by CP, which interferes with LOOH detection in blood samples.

## 1. Introduction

Redox status is a condition that is increasingly investigated in organisms for the evaluation of either physiological or pathological mechanisms. Redox homeostasis is the result of a delicate balance between the production of oxidants and antioxidants. Reactive oxidants, such as reactive oxygen and nitrogen species (ROS and RNS), when produced in small amounts as by-products of aerobic metabolism, play a pivotal role in many physiological functions, regulating many signaling pathways and activating the adaptation and protection behaviors of organisms under stress [1,2]. In this case, the potential harmful effects of these compounds are adequately counteracted by the antioxidant defense system [3,4]. When oxidant production exceeds cellular antioxidant capacity, organisms can experience oxidative stress (OS), with failure in the prevention and repair of oxidative damage to macromolecules that, in turn, affects their structure, causing dysfunction of their physiological activities [5,6].

Most ROS analytical assessments are complex and impractical for clinical or screening applications [7]. Furthermore, since ROS are unstable molecules, their quantification can lead to misleading results [8]. Some alternative analytical assays have been developed that are mainly based on evaluating lipid hydroperoxide (LOOH) content. LOOHs are unstable molecules and the primary oxidation products of polyunsaturated fatty acids (PUFAs); they generate new peroxyl and alkoxy radicals and decompose to form secondary products [9]. LOOH detection methods have been largely applied in both human and animal blood samples and are also available as commercial kits [10,11]. They are mainly based on two different methods: oxygen metabolites (ROMs) [12] and total oxidant status (TOS) [13]. Both methods are based on iron-mediated mechanisms, and the oxidant potential of a sample is measured as the content of substances able to oxidize a chemical compound in the test solution. However, their detection mechanisms are quite different. In fact, the ROMs test assumes that LOOHs generate alkoxyl (R-O^●^) and peroxyl (R-OO^●^) radicals following the release of iron ions from metalloproteins in an acidic buffered solution (Fenton reaction). These radicals oxidize an alkyl-substituted aromatic amine (N,N-diethyl-para-phenylenediamine, DEPPD), generating a pink-colored derivative, which is spectrophotometrically measured at 530 nm [12]. The TOS assay instead evaluates the content of oxidant molecules in a sample through the oxidation of ferrous (Fe^2+^)-*o*-dianisidine dihydrochloride complex to ferric (Fe^3+^) ions. The ferric ions form a colored complex with the chromogen xylenol orange in an acidic medium, which is spectrophotometrically measured at 550 nm [13].

Although both tests evaluate the total LOOH content in plasma or serum, when applied to the same samples, they return non-comparable and unrelated results in both human [13,14] and animal [15] analyses. Previous papers even reported LOOH values that were 100 times higher in goat kid sera using the ROMs test with respect to the TOS test [15,16]. Similar results have been obtained in human samples [13], in which the LOOH levels assessed using the ROMs test largely exceed the cytotoxicity threshold level [17].

The reason for the discrepancy between these two methods is likely due to the use, in the ROMs method, of the alkylamine chromogen DEPPD, which is also a substrate for the enzymatic analysis of ceruloplasmin (CP) [18]. CP is a glycoprotein and the major copper-carrying protein in the blood, and is also involved in iron metabolism. It acts as a ferroxidase enzyme by oxidizing ferrous to ferric ions with the reduction of molecular oxygen to water. The reactive ferrous ions are, therefore, unavailable to catalyze the decomposition of hydrogen peroxide to produce hydroxyl radicals [19]. Thus, CP may be considered an antioxidant. It is also a potent physiologic inhibitor of myeloperoxidase [20], an oxidative neutrophil enzyme responsible for protein and lipid modifications, as well as for the increase in LOOHs during inflammation [21]. Several studies found a strong positive correlation (*p* < 0.001) between ROMs-evaluated LOOH content and CP oxidase activity in both human [13] and animal [15] samples. This finding suggested low specificity of the ROMs test since part of its analytical value may be attributed to substances other than the OS-produced metabolites. In line with this, the addition of sodium azide, an effective CP inhibitor, to human serum extinguishes any relationship between ROMs and CP oxidase activity, and significantly decreases the analytical signal of the ROMs assay [13]. Additionally, Kilk et al. [18] showed a progressive reduction in the absorbance values of the ROMs assay when increasing concentrations of sodium azide were added to human and bovine sera. A reduction in the analytical signal was also observed when pure solutions of human and bovine CP replaced the serum samples, suggesting that the ROMs test mainly measures the CP oxidase activity, and that LOOHs work as potential interfering molecules. Moreover, these authors found that sodium azide did not inhibit CP oxidase activity in chicken and wild redpoll sera, highlighting that LOOHs’ contribution to the test outcome in avian species is higher than in mammals [18].

This study aimed to validate the ROMs test’s effectiveness in detecting the oxidant status of serum samples collected from healthy young small ruminants. The choice to use young animals for this study was based on the need to reduce the possible sources of oxidation, which is more frequent in adult animals and is associated with the rearing system, production/reproductive conditions, and a greater occurrence of diseases, even if this is not evident [22]. To achieve this goal, the ROMs test was compared to either a TOS test or a CP oxidase activity assay. To discriminate the LOOH analytical signal from any CP interference, serum samples were further treated with increasing molar concentrations of sodium azide.

## 2. Materials and Methods

### 2.1. Animals and Blood Sampling

Blood samples were collected from sixteen Camosciata delle Alpi goat kids and sixteen Italian Merino-derived lambs that were randomly chosen from a private farm in the Potenza district (Italy). Pre-weaning goat kids and lambs were aged about 60 and 40 days and weighed 12.8 ± 0.7 and 14.2 ± 1.3 kg, respectively. To deepen and confirm the results, a new group of animals reared on another private farm in the province of Potenza was enrolled for a second trial. This group consisted of Grigia Lucana kids (n = 5) and Italian Merino-derived lambs (n = 5), as well as Grigia Lucana goats (n = 5) and Italian Merino-derived ewes (n = 5) in lactation. All the animals were clinically healthy and free from internal and external parasites. Evaluation of the animals’ health status was based on rectal temperature, heart rate, respiratory rate, appetite, and fecal consistency. Blood samples were collected from the external jugular vein into tubes without an anticoagulant; after clotting, the sera were obtained via centrifugation (2200× *g* for 10 min at 4 °C) and stored at −80 °C until analyses were performed. All procedures were carried out in strict accordance with the European legislation regarding the protection of animals used for scientific purposes (European Directive 2010/63), as recognized and adopted by Italian law (DL 2014/26). No animal suffered as a consequence of blood sampling.

### 2.2. Analytical Methods

Unless otherwise indicated, all reagents and media were purchased from Sigma-Aldrich (Milan, Italy).

Reactive oxygen metabolites (ROMs) were assessed as described by Alberti et al. [12]. Briefly, 10 µL of each sample was added in duplicate to each well of a microtiter plate and, subsequently, filled with 200 µL of an analytical mixture containing 100 mM acetate buffer solution, pH 4.8, supplemented with 0.37 mM N,N-diethyl-para-phenylendiamine (DEPPD) and 2.8 mM iron (II) sulfate heptahydrate (FeSO_4_·7H_2_O). After incubation (30 min at 37 °C), the optical densities (ODs) were read at 530 nm against a blank, where phosphate buffer saline replaced the sample, using a microplate reader (model 550, BioRad, Segrate, Milan, Italy). The assay was calibrated with *tert*-butyl hydroperoxide (*t*-BHP) and the results were expressed in terms of *t*-BHP equivalents (mM).

Total oxidant status (TOS) was assessed as described by Erel [13]. Briefly, 35 μL of samples were added in duplicate to each well of a microtiter plate and mixed with 225 μL Reagent 1 (150 μM xylenol orange, 140 mM NaCl, and 1.35 M glycerol in 25 mM H_2_SO_4_ solution, pH 1.75) and the ODs were read at 550 nm against a blank (see above) using the microplate reader. After that, 11 μL Reagent 2 (ferrous ion 5 mM and *o*-dianisidine 10 mM in 25 mM H_2_SO_4_ solution) was added to the mixture. After 5 min at 37°C, the ODs were again read at 550 nm. The assay was calibrated with *t*-BHP and the results are expressed in terms of *t*-BHP equivalents (μM).

To discriminate the analytical LOOH signal due to possible CP interference, both tests were repeated on pooled serum samples. Each pool was obtained by mixing, at random, four serum samples of each species, thus generating four distinct samples for each species. Then, each pooled serum sample was treated with increasing (from 1 to 1000 µM) molar concentrations of sodium azide just before performing the assays.

The evaluation of CP was based on its oxidase activity using *o*-dianisidine dihydrochloride (ODD) as a substrate, as described by Schosinsky et al. [23]. Briefly, serum samples were incubated in duplicate at 37 °C in the presence of 7.88 mM ODD in 0.1 M acetate buffer, pH 5.0. The light absorption variation in relation to the color intensity of the sample was measured using a spectrophotometer (SmartSpec 3000 UV/Vis, Bio-Rad, Segrate, Italy) against a blank (see above) at 540 nm after 5 min and 15 min of incubation using 9 M H_2_SO_4_ to stop the enzymatic reaction. The CP oxidase activity was expressed in units per liter (U L^−1^) in terms of consumed substrate and calculated as the difference between the two absorbance values [23].

### 2.3. Statistical Analysis

The analytical data, presented as means ± standard deviation (SD), consisted of the averages of three analyses performed for each parameter. The Kolmogorov–Smirnov test was used to determine the normality of the distribution of the data (*p* > 0.05). Differences between species in the ROMs, TOS, and CP values, and the effect of sodium azide on the LOOH assessment, were analyzed via one-way analysis of variance (ANOVA). Bonferroni pairwise comparison was conducted to discriminate differences in the mean analytical values of the ROMs assay at different sodium azide concentrations in comparison with either the lowest or the highest sodium azide concentrations. Bland–Altman analysis was performed to describe the agreement between the ROMs and TOS assays [24]. In particular, the differences between these two analytical methods (*y*-axis) were plotted against the average of the two analytical method (*x*-axis) values using the open-source software Jamovi (The Jamovi project, Version 2.3.21.0). Linear regression analyses were performed to check for possible correlations among the assessed parameters. *p*-values lower than 0.05 were considered to be statistically significant. All these statistical analyses were performed using SigmaPlot software for Windows (Version 11.0, Systat Software Inc., San Jose, CA, USA). The IC_50_ values, defined as the concentration of sodium azide that caused 50% inhibition of the analytical signal in the ROMs assay, were analyzed using Microsoft Excel software.

## 3. Results

In the ANOVA analysis, there is a significant difference in all the analyzed parameters between the two small ruminant species, as reported in Table 1, with a discrepancy between the two analytical assays for the LOOH content. In fact, in the ROMs test, significantly higher LOOH values were found in goat kids than in lambs, whereas a significantly inverse result was found with the use of the TOS test. CP oxidase activity was more than two times higher in kids than in lambs.

The regression analyses did not detect any significant correlations between the ROMs and TOS values in either goat kid (r^2^ = 0.011, *p* = 0.696) or lamb (r^2^ = 0.150, *p* = 0.161) samples, or between the TOS and CP oxidase activity values in either goat kid (r^2^ = 0.059, *p* = 0.363) or lamb (r^2^ = 0.180, *p* = 0.143) samples. Conversely, ROMs and CP oxidase activity levels were highly correlated in both kid and lamb samples (Figure 1).

The poor agreement between these two different assays, both measuring LOOH content, was assessed using the Bland–Altman plot, which shows that the difference between the two methods tends to become larger as the values increase (Figure 2). In addition, all differences, except one, fell within the limits of agreement (mean ± 1.96 standard deviation).

The analysis of the data of the second trial (Table 2) that was carried out on a smaller number of animals and, in the case of goats, on different breeds, confirms the differences in the ROMs, TOS, and CP values between kids and lambs, although the comparison does not allow us to express significant differences due to the high individual variability and the small number of animals used. Upon comparing the data according to age, regardless of the species, adults showed lower levels of ROMs than juvenile animals (0.384 ± 0.143 vs. 0.679 ± 0.151 mM, *p* < 0.001), whereas upon comparing the data without considering age, there is a higher level of TOS in the ovine than caprine species (34.30 ± 19.57 vs. 15.45 ± 6.15 µM, *p* < 0.01).

When sodium azide was added to the samples before the execution of the two LOOH assays, the ROMs test values showed a significant progressive decrease with increasing micromolar concentrations of sodium azide (Figure 3) in both goat kid (Figure 3A) and lamb (Figure 3C) sera (*p* < 0.001), whereas no changes were found in the TOS values following the same treatment (Figure 3B,D). The ROMs values were significantly affected by sodium azide at the lowest concentrations of 31.25 µM (*p* < 0.001) and 7.81 µM (*p* = 0.004) in goat kids (Figure 3A) and lambs (Figure 3C), respectively. On the other hand, the concentrations of sodium azide that ensured the highest reduction in the analytical signal were 250 µM (*p* < 0.001) and 125 µM (*p* = 0.003) in the goat kid (Figure 3A) and in lamb (Figure 3C) samples, respectively. Finally, the IC_50_ values of sodium azide were 98.51 µM and 72.91 µM in the goat kid and lamb samples, respectively. The same treatment with sodium azide also produced results similar to those observed in juveniles when conducted on blood serum from adult goats and sheep (Appendix A).

## 4. Discussion

In this study, LOOH levels, as a biomarker of oxidative status, were detected in the blood sera of both adult and juvenile small ruminants using two different methods: ROMs and TOS assays. In addition, in the same blood samples, CP oxidase activity was also evaluated. The analytical results obtained for these three assays are in line with those reported by other authors, as is the large variability that was found among individuals in their analytical data [15,25,26,27,28,29,30,31]. The differences that emerged between species, as detected in the first experimental dataset and confirmed by the data of the second trial, to which was also added a significant difference between ages, are not supported by the literature due to a lack of these types of comparative analyses between these species. The ROMs and TOS values were not significantly correlated in either goat kid or lamb sera, although both methods were designed to assess the LOOH levels, confirming results previously found in both human and goat serum samples [13,14,15]. Moreover, the poor agreement that emerged in the Bland–Altman plot indicates the existence of proportional bias and, hence, the two methods did not agree equally throughout the range of measurements.

A highly significant correlation was, however, found between ROMs and CP oxidase activity levels in both goat kid and lamb sera (*p* < 0.001). This finding is unexpected considering the role played by CP as a strong inhibitor of myeloperoxidase (MPO) [20]. In fact, since MPO is associated with an increase in LOOHs during inflammation [21], CP, as an MPO inhibitor, should decrease LOOH and, hence, ROMs and TOS levels [13]. In line with this, previous studies [15] detected a high positive correlation (*p* < 0.001) between TOS and MPO activity in the blood serum of goats kid. Moreover, in nitrosative stress, stable end-products of nitric oxide radicals (NO^•^), such as NO_x_, were found to be positively correlated with TOS values in goat kids [15], as well as in cattle [32] and humans [33,34]. This finding is of particular interest as the simultaneous evaluation of NO_x_ and MPO activity can provide important indications of the role played by OS in human disorders [35,36]. Conversely, ROMs values, despite representing alternative assessments of LOOH content [12], were not found to be associated with oxidant and antioxidant biomarkers in human and goat sera [14,15], but only with the CP oxidase activity, as found in the present results and in other studies [13,15]. The relationship between ROMs and CP oxidase activity may bias the ROMs analytical method, making it poorly specific for LOOH assessment. In fact, the ROMs assay returns unrealistic LOOH levels in some mammalian species, which is attributable to analytical interference with CP and other serum components [13,18]. This assumption was demonstrated using either pure solutions of human and bovine CP or human and bovine sera that were treated with sodium azide, a strong CP inhibitor [13,18]. Based on these findings, Kilk et al. [18] speculated that a ROMs assay would mainly detect CP with potential interferences from LOOHs, iron level, thiols, and albumin in human and bovine samples. On the other hand, in some avian species, the ROMs assay represents a method of choice for the evaluation of LOOHs considering that in these species, CP does not seem to interfere with the ROMs assay, as demonstrated by the absence of an effect of sodium azide treatment on the outcomes of the analytical ROMs assay [18]. Therefore, a ROMs assay cannot discriminate between LOOHs and CP, biasing the results, at least in human and bovine serum samples [13,18]. However, Colombini et al. [37] did not find any significant correlations between ROMs and CP values or copper content in human sera, and thus, supported the analytical specificity of the ROMs assay. These conflicting results require further studies aimed at definitively clarifying the reliability of the ROMs test for the evaluation of the oxidative status of organic matrices.

To further support potential interference of the CP with the analytical results of a ROMs assay, however, different pH conditions are foreseen by the two methods under examination. Generally, pH variation has a marked effect on the rate of the enzymatic reaction; in fact, pH alters the charge of functional residues in substrate binding or in the catalysis process itself. Additionally, enzymes can also undergo changes in their conformation, together with changes in pH [38]. The reduction in CP oxidase activity at low pH may be due to the effect of the pH of the analytical mixture in ionic groups on the active site, or variation in the ionic state of the substrate or enzyme–substrate complex [39]. The optimum pH for CP oxidative activity assessment depends not only on the nature of the chemical substrate, but also on the different animal species. Thus, for example, in humans, the optimal pH is 5.4, but at pH 4.6, enzymatic activity is already inhibited by 50–80% [39]. In dogs, the best analytical condition is obtained at pH 5.2, whereas at pH 4.8, CP oxidase activity decreases by about 10–20% [40]. In pigs, Martínez-Subiela et al. [41], using *o*-dianisidine dihydrochloride as a substrate, showed that the optimal pH for detecting CP oxidase activity is 4.6 with a progressive decrease up to about zero at pH 4.2. In our study, the pH of the analytical mixtures was 4.8 and <2 in the ROMs and TOS assays, respectively. While at pH conditions foreseen by the ROMs assay, CP oxidase activity should be partially maintained, and at the very low pH of the TOS assay, complete enzymatic deactivation of CP is likely to occur. Hence, the results of the present study confirm, in young small ruminants, the low specificity of the ROMs assay in the detection of LOOH content, as well as the interference of CP in the analytical method.

The different concentrations of LOOH detected using these two methods is also worth reflecting on. While the analytical values of the TOS assay vary around a few µM *t*-BHP equivalents, the ROMs assay returns values about 100 times higher. This agrees with the values recorded in human samples [13], in which the LOOH levels assessed using a ROMs test were 350 times higher than those obtained using a TOS assay, and far beyond the cytotoxicity threshold indicated in the literature [4,17]. At the same time, ROMs-assessed LOOH values in livestock species are remarkably lower than in humans [25,26,28,42], whereas TOS-assessed LOOH values do not greatly differ between livestock species and humans [43,44,45,46]. This discrepancy may be attributable to the higher CP oxidase activity in humans than in animal species [23,47,48,49,50]. To support this hypothesis, our results found that TOS-assessed LOOH values were higher in lambs than in goat kids, whereas ROMs-assessed LOOH content was higher in goat kids than in lambs. Contextually, the CP oxidase activity in kids was about double that in lambs. Thus, the lower detection of LOOH content using the ROMs test in lambs could be attributable to the lower CP interference in the ROMs analysis, which is associated with lower CP plasma concentration in this species. Moreover, the treatment of the serum samples with a progressively increasing sodium azide concentration led to a progressive decrease in the analytical signal, which reached absorbance values close to those of the blank at the highest sodium azide concentration.

Another finding highlighted using the ROMs assay in young small ruminants is the progressive reduction in or eradication of the analytical signal with increasing sodium azide concentration. This demonstrates that, at least in our animal models, the ROMs assay was able to detect only the CP content. In human and bovine serum samples, Kilk and colleagues [18], despite detecting a reduction in the analytical signal due to sodium azide, observed a residual absorbance value, even at the highest sodium azide concentrations. This suggests that the ROMs assay is able to cumulatively detect LOOH and CP content without discriminating between these two molecules. The discrepancy that emerged between the results of the present study and those of Kilk et al. [18] may be attributed, together with species-specificity, to the young age of the animals enrolled in the present study. In fact, young animals usually have a lower oxidative status than adults, with the exception of the neonatal period [22]. Thus, although the ROMs assay was criticized as a simultaneous measurement of LOOH and CP in humans, our results demonstrate that, at least in young small ruminant specimens, it mainly reflects CP oxidase activity.

## 5. Conclusions

Oxidative status, measured as LOOH content, in the blood serum of lambs and goat kids, was assessed using two different methods, reactive oxygen metabolites (ROMs) and total oxidant status (TOS), together with ceruloplasmin (CP) oxidase activity. Although the ROMs and TOS tests are both designed to measure the LOOH content in clinical and experimental practice, and both are based on iron-mediated mechanisms, the present study demonstrated that these tests are not comparable to each other. In this regard, upon treating samples with sodium azide, a potent CP inhibitor, the analytical signal of the ROMs test was almost completely eradicated, but not that of the TOS test; this suggests that CP interfered with the ROMs assessment, as previously found in humans. Therefore, the present findings highlight the low diagnostic specificity of the ROMs test in measuring the LOOH content in blood serum because it mainly detects CP oxidase activity, at least in young small ruminants.

## Figures and Tables

**Figure 1 antioxidants-12-00701-f001:**
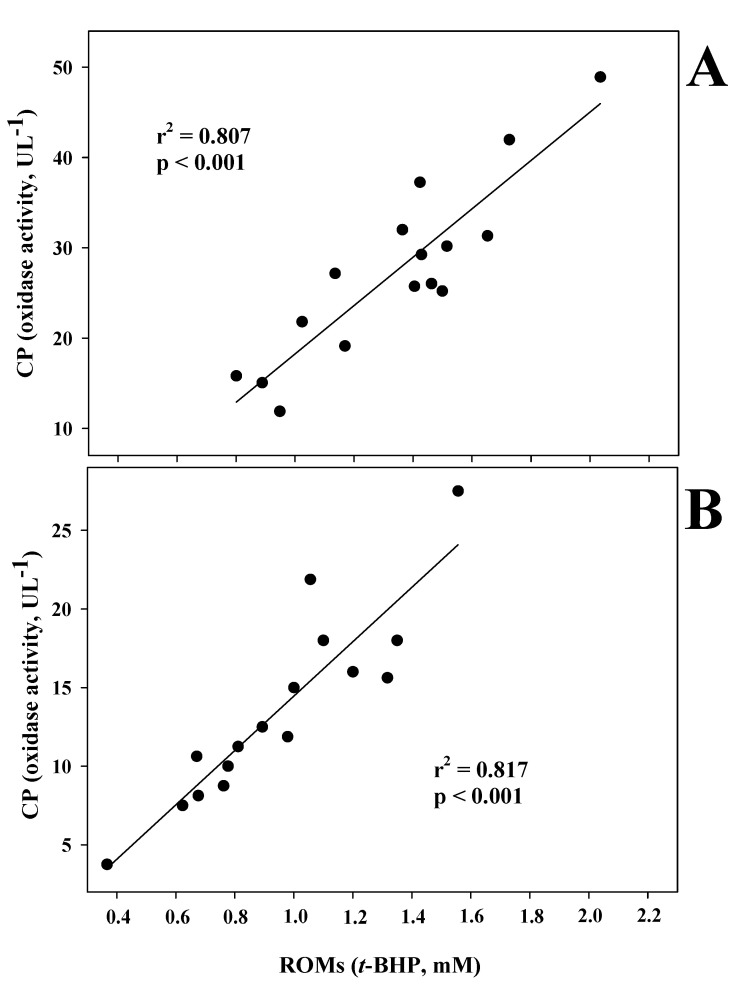
Regression analysis between reactive oxygen metabolites (ROMs) and ceruloplasmin (CP) oxidase activity in the blood serum of goat kids (**A**) and lambs (**B**).

**Figure 2 antioxidants-12-00701-f002:**
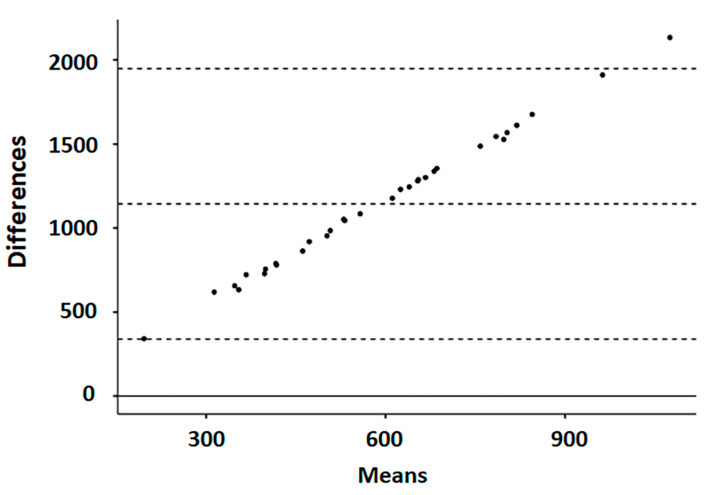
Bland–Altman dispersion plot (average of the two methods against the difference between values) comparing ROMs and TOS assays in small ruminant blood samples. The solid line indicates the height where differences equal to 0 are placed (zero bias line); the central dashed line represents the mean of the differences between the measurements of the two methods (bias); and the two dotted lines at the top and bottom delimit a band that represents the confidence interval limits of the mean of the differences (mean of the differences ± 1.96 × standard deviation).

**Figure 3 antioxidants-12-00701-f003:**
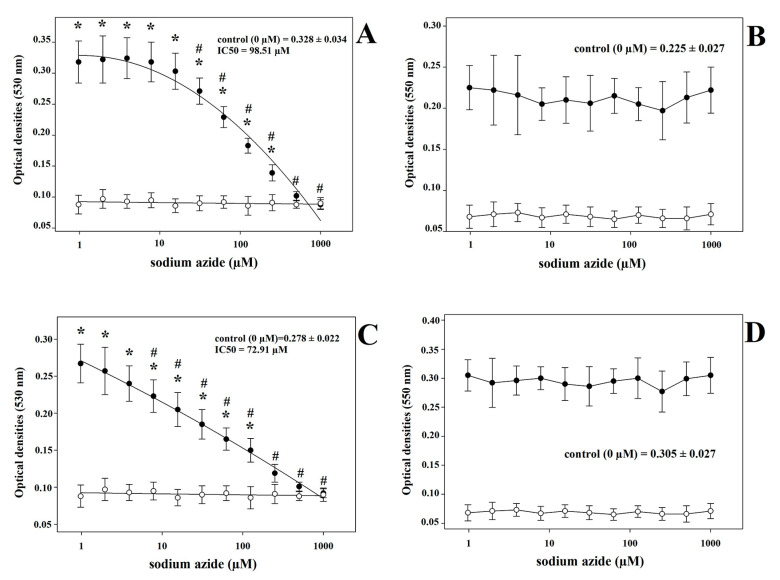
Dose-dependent effects of sodium azide treatment in goat kid (**A**,**B**) and lamb (**C**,**D**) blood serum samples, as assessed using ROMs and TOS assays, respectively. Blank abs: optical densities of blank wells treated with the same sodium azide concentrations as the blood serum samples. Mean values with the superscripts # and * show statistically significant differences in comparison with the lowest (0 µM) and the highest (1000 µM) sodium azide concentrations, respectively.

**Table 1 antioxidants-12-00701-t001:** Mean (±SD) values of the LOOH content evaluated using ROMs and TOS assays, and the ceruloplasmin oxidase activity in blood serum of Comosciata delle Alpi goat kids (n = 16) and Italian Merino-derived lambs (n = 16).

	Goat Kids	Lambs	Significance
	Mean ± SD	Mean ± SD	*p*<
ROMs	1.342 ± 0.330	0.946 ± 0.311	0.01
TOS	13.64 ± 7.68	19.72 ± 9.53	0.05
CP	27.42 ± 9.85	13.52 ± 5.95	0.001

ROMs: reactive oxygen metabolites (*t*-BHP equivalents, mM); TOS: total oxidant status (*t*-BHP equivalents, μM); CP: ceruloplasmin (oxidase activity, U L^−1^); SD: standard deviation.

**Table 2 antioxidants-12-00701-t002:** Mean (±SD) values of the LOOH content evaluated using ROMs and TOS assays and the ceruloplasmin (CP) oxidase activity in blood serum of Grigia lucana goat adults (n = 5) and kids (n = 5), and Italian Merino-derived ewes (n = 5) and lambs (n = 5).

	Goats	Sheep	Goat Kids	Lambs
	Mean ± SD	Mean ± SD	Mean ± SD	Mean ± SD
ROMs	0.286 ± 0.049	0.481 ± 0.139	0.757 ± 0.170	0.601 ± 0.086
TOS	13.46 ± 6.14	29.92 ± 16.12	17.45 ± 6.12	38.69 ± 23.54
CP	5.38 ± 1.44	5.75 ± 2.88	10.75 ± 6.43	5.38 ± 3.14

ROMs: reactive oxygen metabolites (*t*-BHP equivalents, mM); TOS: total oxidant status (*t*-BHP equivalents, μM); CP: ceruloplasmin (oxidase activity, U L^−1^); SD: standard deviation.

## Data Availability

All of the data is contained within the article and the Appendix A.

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
