# Peer review of "Ceruloplasmin Interferes with the Assessment of Blood Lipid Hydroperoxide Content in Small Ruminants"

_antioxidants, 2023, doi:10.3390/antiox12030701_

Round 1

Reviewer 1 Report

The study is interesting and shows how two tests designed to measure the same biological entity in serum instead give results that are neither consistent nor associated. In addition, the study shows in a very simple way the strong interference due to ceruloplasmin activity in the d-ROMS test. The results are very convincing, and the study deserves to be published. It can be further improved though by using the correct statistical test for comparison. actually to compare two tests that measure the same quantitative biological variable the statistical test to use is the Bland-Altman plot. this plot is a scatter plot that allows you to assess the agreement between two quantitative measurements.

It makes sense to use this plot whenever the goal is to understand whether method A's quantitative measurements agree with method B's. Authors can use their data as a whole, without distinguishing between the two animal groups. The comparison is only about the test ROMs vs. TOS. It will be evident that there is a bias. The important difference in the absolute value of the measure between the two tests, when related in the same unit of measurement could in itself prevent the application of the Bland-Altman test. The authors in the case will indicate this as a result. 

in the Bland-Altman test, on the vertical axis are the differences between the two measurements (i.e., measurement error)

On the horizontal axis are the arithmetic averages of the two measurements. Assuming in fact that the two measurements made are equivalent, the best estimate of the true value of the measurement is the arithmetic mean of the two measurements. 

The application of Bland-Altman will strongly corroborate the results presented. 

Author Response

The study is interesting and shows how two tests designed to measure the same biological entity in serum instead give results that are neither consistent nor associated. In addition, the study shows in a very simple way the strong interference due to ceruloplasmin activity in the d-ROMS test. The results are very convincing, and the study deserves to be published.

Reply: We would like to thank the Reviewer for evaluating our manuscript and for raising questions that helped us improve it. We really appreciated her/his valuable comments.

It can be further improved though by using the correct statistical test for comparison.  Actually, to compare two tests that measure the same quantitative biological variable the statistical test to use is the Bland-Altman plot. this plot is a scatter plot that allows you to assess the agreement between two quantitative measurements. It makes sense to use this plot whenever the goal is to understand whether method A's quantitative measurements agree with method B's. Authors can use their data as a whole, without distinguishing between the two animal groups. The comparison is only about the test ROMs vs. TOS. It will be evident that there is a bias. The important difference in the absolute value of the measure between the two tests, when related in the same unit of measurement could in itself prevent the application of the Bland-Altman test. The authors in the case will indicate this as a result. in the Bland-Altman test, on the vertical axis are the differences between the two measurements (i.e., measurement error). On the horizontal axis are the arithmetic averages of the two measurements. Assuming in fact that the two measurements made are equivalent, the best estimate of the true value of the measurement is the arithmetic mean of the two measurements. The application of Bland-Altman will strongly corroborate the results presented.

Reply. We thank a lot the reviewer for her/his reliable advice that we have gladly applied with the aim of improving the quality of our paper. We analyzed our data with the Bland-Altman test as suggested and standardized our data on the same unit of measurement. The Bland-Altman plot shows that the difference between the two methods tends to get larger as the values increase. This indicates the existence of a proportional bias and, hence, the two methods under evaluation do not agree equally through the range of measurements.

Reviewer 2 Report

In this paper the authors compare two methods for assessing blood lipid hydroperoxide content in ruminants and they evaluate the interference by ceruloplasmin. The paper is written clearly but may benefit from a more critical analysis of some points.

1. in the TOS method blood proteins are denatured by H2SO4 so any activity of CP or other enzymes is abolished during the test, this should be clearly stated in the results/discussion 

2. comment on the difference in CP oxidase activity in lambs and goat kids, is it in line with other reports?

minor point:

line 42-43, the sentence is unclear: lipids are organic molecules, what do the authors intend by stating that LOOH are more stable than ROS in organic matrices?

Author Response

In this paper the authors compare two methods for assessing blood lipid hydroperoxide content in ruminants and they evaluate the interference by ceruloplasmin. The paper is written clearly but may benefit from a more critical analysis of some points.

Reply: We would like to thank the Reviewer for evaluating our manuscript and for raising questions that helped us improve it. We really appreciated her/his valuable comments.

  1. in the TOS method blood proteins are denatured by H2SO4 so any activity of CP or other enzymes is abolished during the test, this should be clearly stated in the results/discussion

Reply. We thank the Reviewer for this criticism that prompted us to deepen the study of the mechanisms involved in these two assays, better understand the reasons for their discrepancy, and arrive at a solid explanation. From a careful analysis of the literature, it emerges that the CP oxidase activity is progressively reduced with the lowering of the pH.  However, while the ROMs buffer is pH=4.8, a value associated with a good CP activity, the pH value of the TOS buffer, is lower than 2, corresponding to a nulled activity of the CP oxidase (Martinez-Subiela et al., 2007).

Martinez-Subiela, S., Tecles, F., & Ceron, J. J. (2007). Comparison of two automated spectrophotometric methods for ceruloplasmin measurement in pigs. Research in Veterinary Science83(1), 12-19.

  1. comment on the difference in CP oxidase activity in lambs and goat kids, is it in line with other reports?

Reply. We have included in the Discussion new bibliographic references related to CP oxidase activity as reported by other authors. Our values are in line with those found in the literature. Unfortunately, to our knowledge, there are no comparisons between these two species in relation to these parameters.

minor point:

line 42-43, the sentence is unclear: lipids are organic molecules, what do the authors intend by stating that LOOH are more stable than ROS in organic matrices?

Reply. The reviewer is right: either ROS or LOOH are molecules with a very short half-life. We have remodeled the sentence.

Reviewer 3 Report

The manuscript describes the use of sodium azide as inhibitor of the ceruloplasmin-oxidase activity to infer that this cupro-protein interferes with the measurement of the blood lipid hydroperoxide content, at least with one of the two methods used in clinical for both humans and animals. However, with the data presented, this conclusion is not supported by clear evidences and some major concerns arise.

It is mandatory to determine the specificity of the inhibitory effect of the sodium azide on ceruloplasmin and to exclude any interference of the sodium azide itself with the two methods used.

1) The authors claim in the abstract, in the results and in the conclusions that there isn’t a correlation between TOS and Cp oxidase activity and that sodium azide does not interfere with TOS measurement of LOOH, without showing the data. These results must be included in the manuscript both in figure 1 and figure 2 to support the specificity of the “Cp” interference with ROMs method.

2) Looking the trend of the curve of the inhibitory effect of sodium azide on ROMs method I would expected that above a certain concentration of sodium azide the curve would not decrease anymore since reach the maximum inhibitory effect on the oxidase activity of Cp.

Based on our experience with the human ceruloplasmin, which has a higher functionality than that of the ceruloplasmin of other animals (as also claimed by the authors), about 20 mM sodium azide concentration is able to fully inhibit the ceruloplasmin oxidase activity in human serum. Indeed, the ceruloplasmin concentration in human serum range from 1 to 4 µM which is 5.000 folds lower than the concentration used for the inhibitor (20 mM). Therefore, even if I am not familiar with the concentration of ceruloplasmin in the serum of lambs and goat kids, I would expect that an inhibitory effect specific for Cp occurred at a lower concentrations of sodium azide than those reported and that a residual signal associated to LOOH detection would be detectable even at higher concentrations of sodium azide. Exactly like the scenario previously reported by other investigators that was quoted in the discussion (Kilk et al. ref 17).

The Authors said that the absence of residual detectable LOOH in the lambs and goat kids sera can be due to the absent/very low oxidative status of the serum in young animals and thus ROMs method in these animals only detect the ceruloplasmin activity. This can be demonstrated using the serum of adult animals in which the LOOH must be detectable by the ROMs method also in the presence of sodium azide inhibition of ceruloplasmin activity. Differently, any conclusion can’t be drawn.

Moreover, the claim on the absence of oxidation in the serum of the young animals is contradicted by the quantification of LOOH performed with TOS method, which is considered more trustable by the Authors.

In the light of this reasoning, it seems that an interference of the sodium azide with the ROMs assay occurs, and this must be clarified. I suggest to perform a titration of sodium azide concentration in ROMs assay in the absence of serum samples, to rule out any interference.

3) The conclusion that ceruloplasmin interferes with LOOH detection in ROMs method based just on the use of an inhibitor is weak and more evidences must be shown. For example, a Cp-immunodepletion from sera can be done using polyclonal antibodies anti-ceruloplasmin to demonstrate the direct effect. If specific tools are not commercially available, I am convinced that the use of an anti-human Cp it would work since by quick blast sequence analysis 85% identity and 90% homology occur between human and sheep/goat sequence of ceruloplasmin. Moreover, the use of serum from Cp-KO mice (or other animals if available) would be the perfect control to rule out the Cp interference on ROMs method.

Author Response

The manuscript describes the use of sodium azide as inhibitor of the ceruloplasmin-oxidase activity to infer that this cupro-protein interferes with the measurement of the blood lipid hydroperoxide content, at least with one of the two methods used in clinical for both humans and animals. However, with the data presented, this conclusion is not supported by clear evidences and some major concerns arise. It is mandatory to determine the specificity of the inhibitory effect of the sodium azide on ceruloplasmin and to exclude any interference of the sodium azide itself with the two methods used.

Reply.  We would like to thank the Reviewer for evaluating our Manuscript and for raising questions that helped us improve it. We really appreciated her/his valuable comments. We have deepened the bibliographic analysis and identified solid elements capable of discriminating the two methods analyzed as well as we conducted new experiments to solve some critical issues raised.

1) The authors claim in the abstract, in the results and in the conclusions that there isn’t a correlation between TOS and Cp oxidase activity and that sodium azide does not interfere with TOS measurement of LOOH, without showing the data. These results must be included in the manuscript both in figure 1 and figure 2 to support the specificity of the “Cp” interference with ROMs method.

Reply. We added the info required (Figures 2B and 2D).

 2) Looking the trend of the curve of the inhibitory effect of sodium azide on ROMs method I would expected that above a certain concentration of sodium azide the curve would not decrease anymore since reach the maximum inhibitory effect on the oxidase activity of Cp. Based on our experience with the human ceruloplasmin, which has a higher functionality than that of the ceruloplasmin of other animals (as also claimed by the authors), about 20 mM sodium azide concentration is able to fully inhibit the ceruloplasmin oxidase activity in human serum. Indeed, the ceruloplasmin concentration in human serum range from 1 to 4 µM which is 5.000 folds lower than the concentration used for the inhibitor (20 mM). Therefore, even if I am not familiar with the concentration of ceruloplasmin in the serum of lambs and goat kids, I would expect that an inhibitory effect specific for Cp occurred at a lower concentrations of sodium azide than those reported and that a residual signal associated to LOOH detection would be detectable even at higher concentrations of sodium azide. Exactly like the scenario previously reported by other investigators that was quoted in the discussion (Kilk et al. ref 17).

Reply. We are truly dismayed to have misrepresented the wrong unit of measurement in our paper. An error related to the difficulty of inserting the symbol character occurred in all parts of the manuscript. We are really sorry for this error. Naturally, we dealt with increasing amounts of µM sodium azide rather than mM. As reported by the reviewer, there is no quantitative correspondence between the concentration of ceruloplasmin and the inhibitory effect of sodium azide, in humans, as also referred by Kilk et al., 2014. In particular, Kilk and colleagues found that using ROMs test a 4 µM solution of purified human and bovine CP was inhibited in the presence of various concentrations of azide. Reporting the Authors’ words “The IC50 values of azide for hCP and for bovine CP were 273 µM (95% CI 60–1244) and 124 µM (95% CI 25–611), respectively. Results obtained from human and bovine serum samples were comparable with those obtained with the commercial CP”. In our study, the IC50 recorded in goat kid and lamb blood sera were 98.51 and 72.91 µM, respectively.

The Authors said that the absence of residual detectable LOOH in the lambs and goat kids sera can be due to the absent/very low oxidative status of the serum in young animals and thus ROMs method in these animals only detect the ceruloplasmin activity. This can be demonstrated using the serum of adult animals in which the LOOH must be detectable by the ROMs method also in the presence of sodium azide inhibition of ceruloplasmin activity. Differently, any conclusion can’t be drawn.

Reply. We followed the reviewer's suggestion and set up a new experiment using both adult and juvenile sheep and goat blood sera. The new results are shown in Table 2 and Figure S1. As required, we added the same Na azide concentrations also in adult sera.

Moreover, the claim on the absence of oxidation in the serum of the young animals is contradicted by the quantification of LOOH performed with TOS method, which is considered more trustable by the Authors. In the light of this reasoning, it seems that an interference of the sodium azide with the ROMs assay occurs, and this must be clarified. I suggest to perform a titration of sodium azide concentration in ROMs assay in the absence of serum samples, to rule out any interference.

Reply.  As required, we performed a titration of sodium azide concentration in either ROMs or TOS assays in the absence of serum samples. We reported these results in Figure 3.

3) The conclusion that ceruloplasmin interferes with LOOH detection in ROMs method based just on the use of an inhibitor is weak and more evidences must be shown. For example, a Cp-immunodepletion from sera can be done using polyclonal antibodies anti-ceruloplasmin to demonstrate the direct effect. If specific tools are not commercially available, I am convinced that the use of an anti-human Cp it would work since by quick blast sequence analysis 85% identity and 90% homology occur between human and sheep/goat sequence of ceruloplasmin. Moreover, the use of serum from Cp-KO mice (or other animals if available) would be the perfect control to rule out the Cp interference on ROMs method.

Reply. Due to the time required for the paper's revision and considering the technical time required for getting the antibodies and the necessary checks of the antibody functionality, we could not perform the new experiments as suggested by the reviewer. Anyway, this critique gave us the impetus to deepen the information on the activity of the CP and to carry out future researches on this matter.

However, from the literature analysis, it clearly emerges that the CP activity, as all enzymatic activities, is pH dependent. In children, the maximum enzyme activity has been found at pH 5.4 in partially purified ceruloplasmin (Mahdi et al., 2015). In previous studies in humans, CP oxidase activity showed a maximum at pH 6.0 with about a 15% decrease at pH 5.5 (Henry et al., 1960). Up to date, no studies are available on CP activity at a pH lower than 4.2.  However, in pigs, Martínez-Subiela et al. 2007, using o-dianisidine dihydrochloride as substrate, showed that the optimal pH for detecting CP oxidase activity is 4.6 with a progressive decrease up to about zero with pH 4.2.  Since the optimal activity is species-dependent but grossly ranges between pH 5 and 6 and that this activity presents a bell-shaped curve which probably (not experimentally verified) extinguishes at values < 4, we point out that the ROMs buffer pH is 4.80 whereas TOS buffer pH is lower than 2. For these reasons, we can reasonably report that the TOS method effectively excludes the interference of the CP contrary to what would happen for the ROMs method. We commented on this in the Discussion.

Henry, R. J., Chiamori, N., Jacobs, S. L., & Segalove, M. (1960). Determination of ceruloplasmin oxidase in serum. Proceedings of the Society for Experimental Biology and Medicine, 104(4), 620-624.

Round 2

Reviewer 1 Report

the authors successfully performed the additional tests suggested and the results obtained complete the study. I see only one typo at line 245, Blond-Altman, to be replaced with Bland-Altman

Author Response

Thanks a lot, we fixed the typo.

Best regards

Reviewer 3 Report

The Authors performed the required experiments but in the light of the new results on adult animals the lack of originality/novelty of the message of the manuscript is further underlined.

Indeed, from the inhibitory experiments performed with NaN3 on the serum of adult animals (supplemental materials Figure 1) it is clear that ROMs assay is able to only detect the activity of the ceruloplasmin and not the LOOH content in the blood, being the full inhibitory effect of the NaN3 superimposable to those of the small ruminants. Thus, the interference has the same effect regardless the age of the animals.

However, the interference of ceruloplasmin activity with ROMs assay is already known, as reported by the Authors in the introduction, included the correlation between LOOH evaluated by ROMs assay and ceruloplasmin activity in goat kids (ref 15).

In the light of these evidences also the title of the manuscript is misleading because ceruloplasmin interferes in the assessment of blood LOOH exclusively when measure with ROMs assay and not in general, and is not a prerogative of the small ruminants. Title should be changed.

I am not aware on the relevance of this notion in the field of rearing, for example if some clinical evaluation of the animals status is routinely done with this assay.

I suggest to point towards a “take home message” of the manuscript that is confirmatory of the previously reported observations on the interference of ceruloplasmin on the ROMs assay and that raises a strong “note of caution” for the use of ROMs assay in order to measure LOOH.

This should be introduced in the abstract and in the discussion sections.

Author Response

The Authors performed the required experiments but in the light of the new results on adult animals the lack of originality/novelty of the message of the manuscript is further underlined.  Indeed, from the inhibitory experiments performed with NaN3 on the serum of adult animals (supplemental materials Figure 1) it is clear that ROMs assay is able to only detect the activity of the ceruloplasmin and not the LOOH content in the blood, being the full inhibitory effect of the NaN3 superimposable to those of the small ruminants. Thus, the interference has the same effect regardless the age of the animals. However, the interference of ceruloplasmin activity with ROMs assay is already known, as reported by the Authors in the introduction, included the correlation between LOOH evaluated by ROMs assay and ceruloplasmin activity in goat kids (ref 15).

Reply. We disagree with the reviewer's reflections and believe in the value of our research. Similar research has been conducted in humans and with few cases in cattle. However, (i) the presence of conflicting results in the literature; (ii) the use of two species in which data on the use of these techniques are very scarce; (iii) the comparison between young and adult individuals make this research valid both to support and clarify the use of widely used methods and to introduce new elements of reflection based on solid case studies.

In the light of these evidences also the title of the manuscript is misleading because ceruloplasmin interferes in the assessment of blood LOOH exclusively when measure with ROMs assay and not in general, and is not a prerogative of the small ruminants. Title should be changed.

Reply. In the previous study (Cecchini and Fazio, 2021), the aim was to evaluate the antioxidant status of the blood serum of kids using several techniques. However, we realized that something was wrong when the two techniques evaluating the LOOH content provided different and uncorrelated data. We, therefore, undertook a new study to verify this discrepancy also because we found conflicting data in the literature on a possible interference of ceruloplasmin in the analytical data of ROMs assay.

I am not aware on the relevance of this notion in the field of rearing, for example if some clinical evaluation of the animals status is routinely done with this assay. I suggest to point towards a “take home message” of the manuscript that is confirmatory of the previously reported observations on the interference of ceruloplasmin on the ROMs assay and that raises a strong “note of caution” for the use of ROMs assay in order to measure LOOH. This should be introduced in the abstract and in the discussion sections.

Reply. We believe that the message is strong and clear also without modifying the text or the title;  in addition, in the abstract, we have reached the maximum number of characters allowed and, in the Discussion, we reported exactly this message.